# Age-Related Change in the Association Choices of Two Species of Juvenile Flamingos

**DOI:** 10.3390/ani13162623

**Published:** 2023-08-14

**Authors:** Abbie Loader, Paul Rose

**Affiliations:** 1University Centre Sparsholt, Sparsholt College Hampshire, Sparsholt, Winchester SO21 2NF, UK; abbieloader@uwclub.net; 2Cotswold Wildlife Park, Bradwell Grove, Burford OX18 4JP, UK; 3Centre for Research in Animal Behaviour, Psychology, University of Exeter, Perry Road, Exeter EX4 4QG, UK; 4WWT Slimbridge Wetland Centre, Slimbridge, Gloucestershire GL2 7BT, UK

**Keywords:** flamingo, association, behavioural development, enclosure use, animal husbandry, animal welfare

## Abstract

**Simple Summary:**

Flamingos are highly social birds that live in large flocks. Flamingos are commonly housed in zoological collections but can be challenging to breed regularly. The characteristic bright pink plumage colour of adult birds develops over time. Anecdotally, young flamingos have been observed to appear on the edges of flamingo flocks. Therefore, this project aimed to measure the social behaviour of juvenile flamingos and record where they were found in two zoo enclosures for two species of flamingo. This project identified that juvenile flamingos prefer to spend time with birds of their own age and are most commonly seen on the periphery of their flock. However, this can be dependent on species and situation (e.g., management impacts on flock structure). We suggest improvements to flamingo enclosures to provide quality, accessible resources for all individuals of all ages that are housed under human care.

**Abstract:**

Flamingos are colonial species commonly kept in zoos, well known for their bright plumage and elaborate courtship displays. This project aimed to determine the differences in flock position and association preferences of juvenile Greater Flamingos (*Phoenicopterus roseus*) and Caribbean Flamingos (*P. ruber*) housed in the same zoological collection. Little research has been conducted on the association preferences of juvenile flamingos, especially in captive flocks, and therefore this study collected data using photographs taken throughout 2014 and 2015 to further understand association patterns. Data were collected on the age category of each juvenile flamingo observed, the age of their nearest neighbour and their position within the flock, and the location within an enclosure zone at different times of the day. The results showed that Greater Flamingo juveniles mainly associated with individuals of their own age and were most likely positioned at the periphery of their flock significantly more of the time until approximately 24 months of age. Sub-adult Greater Flamingos spent significantly more time associating with adult flamingos at the centre of the flock. In contrast, data collected on Caribbean Flamingos indicated that juveniles did not segregate themselves from the adults as distinctively. Birds aged 13–24 months were observed significantly more at the centre of the flock and had more associations with adult flamingos, in a similar manner to that observed in Greater Flamingos. Due to population management needs, juvenile Caribbean Flamingos were removed from the flock at the start of 2015 and this may have influenced the association and location preferences of the remaining young flamingos. In conclusion, these results indicated that captive juvenile flamingos were often seen away from adult birds and that sub-adult flamingos returned to the heart of their natal flock to associate significantly more with other adult individuals, potentially preparing for mate selection and breeding. Captive enclosure should therefore be spacious enough to enable young flamingos to remove themselves from adult birds so that behavioural development can be unaffected by artificially high rates of aggression.

## 1. Introduction

Facilitated by frequent interactions, social animals form complex relationships with their surrounding conspecifics to create an intricate and dynamic social structure within their group. The intricacy and complexity of these existing relationships within a population can make social integration challenging for all newcomers, including juveniles [1]. Across many vertebrate taxa, it is common for newly born or hatched individuals to spend a period in isolation from the group, fully dependent and solely interacting with parental individuals before later starting to interact and develop social relationships with others [2]. These early interactions with new individuals are crucial to behavioural development and social integration [3]. Individuals that lacked social interaction at an early age rarely thrive within a group later in life, and often show impaired reproductive output and health, and reduced longevity [4].

In social birds, the transition from a parental or family unit into the wider group is often quite sudden [5] and it is common for young animals to be abruptly cut off from parental support [5,6]. Both environmental and social pressures cause the development of juvenile-centric groups; across many different bird species, from vultures (Accipitriformes) [7,8] to gulls (Charadriiformes) [9] to flamingos (Phoenicopteriformes) [10], juvenile social associations occur because of dispersal patterns and/or pressures from the presence of mature adults and resulting increases in aggression. Juvenile associations can also favour the development of behaviours with adaptive fitness benefits, as in the case of foraging choices in groups of juvenile Hihi (*Notiomystis cincta*) [11].

During this period of adolescence, many bird species choose to interact with conspecifics their own age, particularly siblings [12]. Free-ranging juvenile Greylag Geese (*Anser anser*) will aggregate in small sibling-only groups, particularly during their first breeding season, and are shown to have higher social connectivity during their first year compared with that demonstrated by birds aged two to three years [13]. This increased social connectivity is likely due to the older individuals spending more time with a single partner after attainment of sexual maturity. Juvenile geese with high levels of social connectivity were also more likely to attempt to breed at a younger age, and successful breeders had a higher number of fledged goslings [13,14]. This research on geese suggests a sudden change in the sociality and social standing of fully matured individuals; during adolescence, the quantity of social bonds appears to be a priority, but these are primarily restricted to interacting with other juveniles whereas in mature individuals, regardless of age, the focus is on courtship and breeding attempts.

Similarly, social integration during adolescence for captive juvenile Gouldian Finches (*Erythrura gouldiae*) showed birds to be directing most social interactions towards siblings, followed by unrelated peers of the same age, then to mature females, and least frequently towards mature males [15]. This led to the creation of temporary, adolescent subgroups within the flock, often comprising of related individuals [15]. These results once again highlight a period in which immature birds are somewhat segregated from the main group, regardless of relation. Whilst they are unable to integrate with mature individuals, young birds are reliant on each other to begin learning social queues and building networks.

In these goose and finch examples, the method of social integration was intentional, with juvenile individuals clearly identifying siblings and peers their own age and assorting themselves into small subgroups. Alternatively, in many birds, a similar process occurs but one which is influenced by environmental factors or physical restrictions; for example, reduced flying ability leads to fledgling wild Pinyon Jays (*Gymnorhinus cyanocephalus*) grouping together whilst parents come and go [16,17]. In some cases, juveniles are constrained by physical or morphological differences and are corralled into large aggregations [18]. These chick aggregations, or crèches, are often observed in colonial nesting birds, particularly seabirds and waterbirds, including penguins (Sphenisciformes), Common Eiders (*Somateria mollissima*), and flamingos [19]. There appears to be multiple benefits to this crèching behaviour such as reduced predation risk, reduced exposure to aggression from adults, and the creation of nursery systems that allow parents to spend increased time foraging [19,20].

All six species of flamingo display crèching behaviour, where a large number of chicks cluster together, usually supervised by several adult or sub-adult birds [21,22]. Flamingo chicks are considered semi-precocial [23]. At hatching, a chick is covered in white down and is unable to stand; however, by one week of age, it can move around more confidently and may leave the nest, whilst being monitored by one or both parents [24]. During this period, heightened aggression is observed between parent birds and surrounding adults [25]. Up until flamingo chicks join a crèche, at approximately 10–12 days of age, they are fed by both parents in short frequent bouts that reduce in number but increase in duration as the chick ages [24,26]. Once in the crèche, feeding usually begins at dusk and continues into the night, with potentially hundreds and thousands of feeds taking place simultaneously [26]. The adults break up the aggregation of chicks by walking through and vocalising in search of their own youngster. Flamingo chicks are able to recognise their parent from its specific vocalisations and will approach in a slightly hunched posture, begging for food [27]. During this stage, not all chicks are fed daily but the duration of each bout of feeding is prolonged, averaging approximately 15 min [24]. By around 30 days, some chicks may start finding small amounts of food for themselves; however, they still have not developed the lamellae required to adequately filter food. As such, chicks are usually fed up to the point of fledging, and it is extremely rare that a chick will continue to be fed after they leave the crèche [24,28].

Soon after fledging, at around 70 to 100 days of age, juvenile flamingos disperse and may travel large distances from their natal colony [29]. The movement of juvenile flamingos at this age is sporadic and they may visit multiple sites that are suitable for breeding and wintering [24]. The rate of natal dispersal, the process in which individuals move from their birth site to their first breeding site, is very high in flamingos, whereas breeding dispersal, where individuals change site between breeding attempts, is more infrequent [30]. However, a more recent study highlights the process of natal philopatry in flamingos later in life; the rate of breeding site fidelity is higher at the natal colony and increases with the level of breeding experience the bird has, while breeding dispersal is mostly directed toward the natal colony [31], overall suggesting that young flamingos will often gain courtship and breeding experience at non-natal colonies, before returning to their birth site once fully matured [30,31].

Current records suggest that the youngest flamingo observed breeding was a three year old Greater Flamingo [32], and by this age birds have developed a plumage colour that nearly resembles the adult, although they likely still retain some features of juvenile plumage such as darker leg joints [32]. It is worth noting that most wild flamingos are observed first breeding between the ages of four and eight years old, when they have fully matured [24]. Flamingos are most likely seasonally monogamous [33] and therefore they are frequently faced with the task of having to select a mate via a synchronised group courtship display [34]. Both males and females participate in this group courtship ritual [34], which allows for mutual selection [35]. Studies suggest that both the performance of behaviours during the courtship display and plumage colouration are an important part of mate selection [35,36].

The personal observation, by one of the authors of this paper, of captive flamingo flocks for previous research projects suggested that juvenile birds, who have yet to gain their adult plumage, appear to stand more frequently at the periphery of their flock, often slightly segregated from adult birds in full pink plumage. Therefore, the aim of this study was to empirically assess the flock position and association patterns of juvenile flamingos in captivity, where natal dispersal is not an option following fledging, and determine any changes that may occur throughout different developmental stages.

## 2. Materials and Methods

Two captive flamingo flock housed at the Wildfowl & Wetlands Trust (WWT) Slimbridge Wetland Centre were used for this research. One flock of 274 Greater Flamingos (*Phoenicopterus roseus*) was housed in a single flamingo species enclosure but mixed with different species of wildfowl (Anseriformes) and the other flock of 134 Caribbean Flamingos (*P. ruber*) were housed in a single-species exhibit. These two species were chosen for this research project based on their similar anatomy, morphology, ecology, and behaviour [21]. Maximum number of birds observed at any one time from a photograph from each year of data collection is shown in Table 1. Juveniles would mature into sub-adults across the time period of data collection and therefore the number of birds within each age category that could potentially be observed would change accordingly.

Free-living Mallards (*Anas platyrhynchos*), Tufted Ducks (*Aythya fuligula*), Moorhens (*Gallinula chloropus*), Coots (*Fulica atra*), Greylag Geese (*Anser anser*), Mute Swans (*Cygnus olor*), Grey Herons (*Ardae cinerea*), various gulls (Laridae), and pigeons (Columbidae) could be seen within the flamingo enclosures. This study utilised photographs taken for data collection for a larger project that investigated social bonds across all six species of captive flamingo housed by WWT. Further information on this project can be found in [37,38,39,40]. Photographs were taken on a Panasonic Lumix T5 digital camera. Ethical review was provided by the Animal Welfare & Ethics Committee of WWT for the original data collection for the social networks project, and this case study was ethically reviewed by the Ethics Committee of Sparsholt College, Hampshire. Overall, 8680 individual data points (Greater Flamingos—4588; Caribbean—4092) were collected retrospectively from 825 photographs taken between January 2014 and December 2015. For the original data collection, photographs were captured at four times per day, 10:00, 12:00, 15:00, and 16:30 (depending on season and bird housing) and full methodological details for the original research for which these images were captured is available in Rose and Croft [37] and Rose and Croft [39]. Photographs captured the entire enclosure and flock within, as well as close-up views of individual subgroups of birds. From the overall sample of photographs across these two years, only those that contained juvenile flamingos were used for this investigation. This allowed quantification of where flamingos were, based on their location in the whole enclosure photo, and, in comparison, the location and position of other birds around them. The same researcher (AL) collected all data from all photographs. An example of a flock-wide and close-up photograph for each flamingo species is shown in Figure 1.

The date and time of capture was recorded and then, starting from the left-hand side scanning to the right, each visible juvenile was identified, and an approximate age was recorded. For the purpose of this study, the plumage colour development chart published by Johnson et al. [41] was used to record the approximate age, with each individual being categorised into one of three developmental stages (Up to 12 months, 13 to 24 months, and over 24 months). For each juvenile flamingo, the individual closest in proximity was identified and visually aged in the same manner, with an additional age category added for fully mature individuals. Figure 2 provides an illustration of each age category for each species.

Finally, the within-flock position and location within the enclosure of each juvenile flamingo was recorded. For the purpose of this study, juveniles were recorded as being in the periphery of the flock if they were visibly away from the main body of the flock or had less than five other individuals between themselves and the edge of the group. Juveniles were recorded as being centrally located in the flock if they were surrounded by other flamingos with no obvious distance (more than one flamingo body length) between themselves and other birds. Nearest neighbour was determined by the data processor (AL); a ruler was used to measure the gap between the juvenile and birds nearby (measuring where the focal bird’s leg met the body, and the nearest leg from a neighbouring bird). The premise behind this was to represent the gap between where the two flamingos were standing, as opposed to judging the nearest point of the flamingo, this being an outstretched head/wing tip, etc., which would bias the identity of potential neighbours. Judging the depth of the flock and location of a juvenile was based on comparison of multiple photographs; because images had been taken at the same time from different angles, it was possible to identify where a juvenile was in relation to others in the main group. Where birds had more than one nearest neighbour, these were noted as impossible to determine, to be excluded from final analyses. After all data were processed, all records of multiple nearest neighbours were sub-adult flamingos where the two nearest neighbours were adult and therefore records were included in analyses.

Enclosure location was recorded using pre-prepared maps that divided each enclosure into recognisable areas of differing resources. The Greater Flamingos’ enclosure was divided into nine zones and the Caribbean Flamingos’ enclosure was divided into fourteen zones. The Greater Flamingo house was not included as a zone because it was not always accessible to the birds whereas the Caribbean Flamingo had free-choice access to their indoor housing. Enclosure zones for each exhibit, separated by resource (e.g., specific area of a pool) or defined feature (e.g., substrate, planting, or physical structure in the enclosure), are provided in Table 2 and were based on those used for previous study [42]. Enclosure features were not directly comparable, but size of favoured terrestrial space was, i.e., the main terrestrial area used by the Greater Flamingo flock was the island (10% of enclosure), and for the Caribbean Flamingos, the sanded area at the rear of their enclosure (11% of enclosure).

### Data Analysis

All data were analysed using Minitab v.17.3.1 [43] to identify any significant relationship between the age category of juvenile flamingos, the age of their nearest neighbour, and their position within the flock. Prior to statistical analysis, all data were reviewed for normality using the Kolmogorov–Smirnov in Minitab, where *p* values of <0.05 suggested non-normally distributed data. Chi-squared goodness of fit tests (χ^2^) were used to determine significant differences in both the flock position (flock periphery or flock centre) and the age of nearest neighbour for each age category. A two-sample proportions test was then used to further compare the number of observed associations between two similarly aged birds against the number of observed associations of differently aged birds for the whole population and for each age category in turn.

For each flock, the different enclosure zones were combined into pool, land (grass and sand), islands, and feeding zone. Counts of birds within enclosure zones and the percentage of observations of flamingos of each age category and their position within a zone (peripheral or central to the main flock) are presented in tables.

A Poisson regression was used to analyse any influence of month of study on the preferences of flamingos as they aged over the course of the data collection period. For each age category (as defined above), a Poisson regression was run with the count of flamingos associating per observation as the outcome variable, and with the specific age classes of a neighbour, month of observation, time of day (morning, noon, and afternoon), and year of data collection as the predictors. Wald’s Chi-squared tests are provided as outputs from Poisson regression. Year was not included as a predictor for 13–24-month-old Caribbean Flamingos due to the movement of birds from the flock for management purposes.

## 3. Results

### 3.1. Differences in the Flock Position of Flamingos at Different Ages

Juvenile Greater Flamingos up to the age of 24 months spent more time at the periphery of the flock than the centre, whilst individuals over 24 months of age were observed more at the centre (Figure 3). Chi-squared analyses showed a significant difference in all three age categories. Juveniles up to 12 months were observed at the periphery of the flock 1085 times and at the centre 894 (χ^2^ = 18.43; df = 1; *p* < 0.001), 13–24-month-old individuals were observed at the periphery 791 times and at the centre 622 times (χ^2^ = 20.21; df = 1; *p* < 0.001), whilst sub-adults were observed significantly more in the centre of the flock (643 times) compared with at its periphery, 572 times (χ^2^ = 4.15; df = 1; *p* = 0.042).

Little difference in the flock position of juvenile Caribbean Flamingos up to 12 months was noted, whilst individuals aged 13 months and older were observed more at the centre of the flock (Figure 4). Juveniles up to 12 months of age did not show a significant difference in flock position (χ^2^ = 0.21; df = 1; *p* = 0.652), being observed at the periphery of the flock 421 times and at the centre 408 times. However, the two oldest categories showed a significant preference for positioning themselves at the centre of the flock. Individuals aged 13–24 months were observed at the periphery 80 times and the centre 111 times (χ^2^ = 5.03; df = 1; *p* = 0.025) and sub-adults were observed at the periphery 1475 times and the centre 1639 times (χ^2^ = 8.64; df = 1; *p* = 0.003).

Both the differences and similarities in the enclosure zone occupancy for juveniles of each flamingo species are noted (Table 3). The youngest age category of Greater Flamingos was more likely to be peripheral in the feeding area compared with Caribbean Flamingos of the same age group. However, when comparing main terrestrial zones (island for Greater Flamingos and land for Caribbean Flamingos), the same pattern is evidenced. Juveniles become more central to the flock in these zones as they age. Birds in all age categories of both species are more peripheral in pool usage throughout each age category.

### 3.2. Differences in Age of Nearest Neighbour

The number of times juvenile Greater and Caribbean Flamingos, within each category, were observed standing close to another bird of the same or different age group are illustrated by Figure 5 and Figure 6. Greater Flamingos of each age category were most commonly recorded with an associate from their own age group (Figure 5), whereas older Caribbean Flamingos were often seen with an adult bird as well as with others of the same age category (Figure 6).

Analyses revealed that all the Greater Flamingo age categories had significant differences regarding choice of nearest neighbour. Individuals up to one year old had another juvenile of the same age group as their nearest neighbour 1124 times, a juvenile aged 13–24 months 100 times, sub-adults 75 times, and with an adult flamingo 650 times (χ^2^ = 1543.05; df = 3; *p* < 0.001). Juveniles aged 13–24 months of age were observed with an individual aged up to a year as their nearest neighbour 102 times, with another individual aged 13–24 months 696 times, with a sub-adult 117 times, and with an adult 498 times (χ^2^ = 728.58; df = 3; *p* < 0.001). Finally, sub-adults were observed next to a juvenile aged up to one year 70 times, a juvenile aged approximately 13–24 months 111 times, a sub-adult 243 times, and with an adult flamingo 790 times (χ^2^ = 1093.64; df = 3; *p* < 0.001).

Significant results were also found for all three age categories in the Caribbean Flamingo flock. Juveniles up to a year of age had another juvenile aged up to a year as their nearest neighbour 420 times, a sub-adult 125 times, and with an adult 280 times; they were not observed stood with a juvenile aged 13–24 months (χ^2^ = 486.15; df = 3; *p* < 0.001). Those birds aged 13–24 months who were not seen with a juvenile aged less than a year as a nearest neighbour, were observed with another individual their own age 40 times, a sub-adult 58 times, and an adult flamingo 94 times (χ^2^ = 95.5; df = 3, *p* < 0.001). Finally, sub-adult Caribbean Flamingos had a juvenile aged less than a year as their nearest neighbour 123 times, a juvenile aged 13–24 months 32 times, a sub-adult 1240 times, and an adult 1719 times (χ^2^ = 2677.54; df = 3; *p* < 0.001).

### 3.3. Preference to Stand near Other Individuals of a Flamingo’s Own Age

Juvenile Greater Flamingos of up to a year of age were significantly more likely to be seen with an individual their own age as their nearest neighbour (N = 1124; Z = 9.69; *p* < 0.001). These one-year-old birds were observed having a nearest individual of a different age category on 825 occasions. There was no significant difference between juvenile Greater Flamingos aged 13–24 months having a nearest neighbour of the same age category or different (Z = −0.79; *p* = 0.429). These birds were observed near an individual of the same age 696 times and with an individual of a different age 717 times. Sub-adult Greater Flamingos were observed significantly more frequently with individuals of different ages than those their own age (Z = −36.92; *p* < 0.001), as they were observed next to an individual of the same age 243 times and an individual of a different age 971 times.

Juvenile Caribbean Flamingos up to 12 months were observed with a nearest neighbour of their own age 420 times and of a different age 405 times, giving no significant difference overall (Z = 0.74; *p* = 0.460). Juvenile Caribbean Flamingos aged 13–24 months did show a significant preference for standing with another individual from a different age category (Z = −14.07; *p* < 0.001) and were observed with a nearest neighbour of their own age 40 times and a nearest neighbour of a different age 152 times. Sub-adult Caribbean Flamingos were observed significantly more with a nearest neighbour from a different age category (Z = −16.41; *p* < 0.001), with the data showing that a nearest neighbour from their own age category occurred 1240 times and with a bird from a different age group 1874 times. The number of observations for each age category with the same or different nearest neighbour age for each species of flamingo are illustrated by Figure 7 and Figure 8 and both figures show that, as flamingos age, the age of their nearest neighbour becomes more varied.

### 3.4. Changes in Nearest Neighbour throughout the Year

There was a significant relationship between the aging of the juvenile flamingo and their choice of nearest neighbour. For Greater Flamingos aged 0–12 months, there was a significant influence of year (χ = 13.53; df = 1; *p* < 0.001), time of day (χ = 18.68; df = 2; *p* < 0.001), month (χ = 333.3; df = 10; *p* < 0.001), and age category (χ = 428.6; df = 3; *p* < 0.001) on changes in the flamingo’s nearest neighbour’s age over time. For Greater Flamingos aged 13–24 months, there was also a significant influence of year (χ = 22.77; df = 1; *p* < 0.001), time of day (χ = 57.58; df = 2; *p* < 0.001), month (χ = 67.14; df = 10; *p* < 0.001), and age category (χ = 155.46; df = 3; *p* < 0.001) on the change in the age of its nearest neighbour over time. For sub-adult Greater Flamingos (>24 months), there was no significant effect of year on the choice of nearest neighbour (χ = 0.76; df = 1; *p* = 0.383) but the time of day (χ = 12.08; df = 2; *p* = 0.002), month (χ = 66.32; df = 10; *p* < 0.001), and age category of the nearest neighbour choice (χ = 68.76; df = 3; *p* < 0.001) were significant predictors. Across each category of Greater Flamingos, the number of associations was higher at midday and in the afternoon when compared with morning observations.

The choice of nearest neighbour over time for Caribbean Flamingos aged 0–12 months was significantly predicted by year (χ = 34.16; df = 1; *p* < 0.001), time of day (χ = 7.96; df = 2; *p* = 0.019), month (χ = 71.30; df = 7; *p* < 0.001), and age category of nearest neighbour (χ = 103.11; df = 2; *p* < 0.001). As with Greater Flamingos of 0–12 months, so too were more observations of associating Caribbean Flamingos noted at midday and in the afternoon when compared with the morning. For Caribbean Flamingos in the 13–24 month age category for 2014 only, the time of day (χ = 0.53; df = 2; *p* = 0.766) and month (χ = 6.66; df = 6; *p* = 0.354) were not significant predictors of the choice of nearest neighbour; however, there was a significant effect of age class (χ = 6.67; df = 2; *p* = 0.036) with birds significantly less likely to be associating with adults or sub-adults over the course of the observation period. Finally, for sub-adult Caribbean Flamingos (>24 months), there was no significant influence of the time of day on association choices (χ = 4.96; df = 2; *p* = 0.084) but the month (χ = 180.95; df = 11; *p* < 0.001, year (χ = 4.14; df = 1; *p* = 0.042), and age of the nearest neighbour (χ = 80.60; df = 3; *p* < 0.001) were all significant.

Figure 9 (data on 0–12-month-old birds) and Figure 10 (data on >24 month old birds) show the fluctuations in nearest neighbour age category over the two years when plotting observations of the 0–12 month juveniles, the 13–24 month juveniles, and the over-24 month juveniles as the age category of an associate.

### 3.5. Enclosure Usage by Age Category

Table 4 illustrates where juvenile flamingos of different age groups were most likely to be seen assorting with birds of their own and of different age classes. Pool use is higher for older Caribbean Flamingos, showing more assortment with adults in this area. The assortment of younger birds in feeding areas occurs irregularly. Greater Flamingos aged 13–24 months were associating infrequently with other birds on the main terrestrial area of the enclosure (the island) whereas Caribbean Flamingos of this age spent more time on land.

## 4. Discussion

This case study investigated the differences in social choice by age and development in two species of flamingo held in a captive environment. In the case of both species, younger flamingos were more likely to be observed away from, and peripheral to, the main group of adults. Our results show that juvenile Greater Flamingos, up to approximately 24 months of age, can be observed at the periphery of the flock significantly more than in the centre. However, once these Greater Flamingos entered sub-adulthood the reverse effect is true. Similarly, Caribbean Flamingo juveniles aged over 24 months were also observed significantly more at the centre of the flock than at the periphery. Over 24 months of age, sub-adult flamingos were developing a plumage more similar to that of adults than to that worn by younger juvenile birds, and in the wild, they would be visiting non-natal colonies to gain courtship display and breeding experience during this period of maturation [24,30,31]. Although this dispersal period begins when juvenile flamingos can fly and leave their colony [29], our results suggest that juvenile flamingos do not begin to re-immerse themselves into the main body of a flock until much later in their development. Although limited in scope, due to observations occurring only in one zoological facility and on one flock of each species, we have shown how young flamingos change in their choice of social partners and in how they interact with the rest of their flock as they age.

Wild flamingo chick dispersal appears to be dependent on body condition, with chicks in a good body condition more likely to leave their natal area [29]. In a captive environment, body condition may influence how likely a juvenile flamingo is to receive aggression from other birds, and therefore juveniles of a better body condition may be more able to reintegrate into the flock as they age. A facilitated learning hypothesis suggests that delayed maturation may help young animals learn important behavioural skills before entering the world of adulthood [44]. In the case of flamingos, the delay in development of full plumage colouration may allow young birds the time to perfect filter feeding behaviour, utilising resources where adults are feeding but not attracting undue attention; this enables growing flamingos to gain enough energy and to acquire a good body condition to eventually join in with courtship displays and breeding activity. Free-living juvenile Caribbean Flamingos have been documented to receive significantly more aggression from adults birds when feeding and are more likely to be displaced from an occupied foraging patch [10]. Such findings have been noted in other bird species too, e.g., black-headed gulls (*Chroicocephalus ridibundus*) [9]. Social assortment between juveniles without the presence of adults increases foraging efficiency, resulting in improvements to body condition due to a less aggressive social setting. Therefore, flamingo enclosures should provide multiple foraging locations to allow juvenile birds to feed with other flamingos of a similar age in areas away from adults to avoid overt instances of aggression that can impact on food collection.

There was no significant difference in the flock position of Caribbean Flamingos aged less than a year, and juvenile Caribbean Flamingos aged between 13 and 24 months were observed significantly more at the centre of the flock than at the periphery. It is worth noting that, at the start of 2015, juvenile Caribbean Flamingos had to be removed from the flock due to heightened aggression from adult birds. Whilst this reduced the eventual dataset for juveniles aged less than 24 months, these results could potentially indicate ineffective crèching behaviour, causing juveniles to be subjected to aggressive behaviour. Research conducted at Dublin Zoo comparing the number of aggressive behaviours observed in different areas of the Chilean Flamingo (*Phoenicopterus chilensis*) enclosure found the level of aggression within the crèche to be the lowest of all the areas, and around the nest site to be the highest [45]. This Dublin Zoo study further documented that adult flamingos entering the crèche to feed their chick were tolerant of other parent birds but retained a high level of aggression towards non-parent individuals within the crèche. Together, these findings and the observation from our study support Toureno et al. [25] who showed that aggression between adult flamingos could be a key contributor to the formation and development of crèches.

### 4.1. Enclosure Location, Flock Position, and Nearest Neighbour

Greater Flamingos categorised as being less than 12 months old and between 13 and 24 months old were observed next to another individual of their own age group significantly more than any other age category (both individually and combined). During the crèching period, this result may be expected, with all young of the same age moving away from the main body of the flock; however, our results show that these preferred associations continue past fledging. Similarly, Caribbean Flamingos less than 12 months old were seen next to an individual of their own age significantly more than each of the other individual age categories (Table 4), but, overall, there was no significant difference in the association preference of these youngest birds. Juveniles aged between 13 and 24 months were observed with an adult as their nearest neighbour significantly more than any other age group and were more likely to be standing nearest an individual of a different age to themselves. This closer proximity of Caribbean Flamingo juveniles to adult birds could explain why they were subjected to higher levels of aggression within the flock. The defined nesting and crèching area in the Greater Flamingo enclosure (Table 2) may have provided a defined space for chicks to congregate in, away from non-parent adults, thus reducing aggression. Therefore, we recommend that all flamingo exhibits should include a defined nesting and crèching area to encourage chicks to congregate in a safe and secure place, easily accessible by parents and with room for chicks to group together away from non-breeding birds (Figure 11).

In both species, sub-adults were observed with adults as nearest neighbours significantly more than any other age category. Our results show that sub-adults are the only age category to show a clear significance toward standing near individuals of a different age, specifically fully matured individuals (Table 4). Again, there appears to be this drastic change in social choice once the juvenile flamingos reach sub-adulthood, and no longer have the mostly grey plumage of adolescence. This may be due to sub-adult birds wanting to join in with adult flock-wide behaviours, such as courtship displays.

Our results indicate that juvenile flamingos spend significantly more time with individuals of their own age at the periphery of the flock, until they enter sub-adulthood somewhere after 24 months of age, at which point they move toward the centre of the flock and focus on associating with fully adult birds. This suggests a small-scale version of the natal dispersal and philopatry mentioned in previous studies [31,46]. In wild Greater Flamingo flocks, younger, more inexperienced, breeders struggle to compete for mates and nesting sites, especially within larger colonies [31]. As such, it may be necessary for the juveniles to breed at another location and return to their natal colony once they have gained experience and are better equipped to compete for choice nesting locations within a colony. In captivity, it is not possible for young flamingos to leave and return once more experience has been gained, although our results seem to suggest attempts at dispersal and return to the main body of the flock but on a smaller scale. The increased association with sexually mature individuals in the later stages of development suggests a period of preparation for sexual selection and reproduction. Therefore, flamingo husbandry and management needs to provide juvenile and sub-adult birds a time away from the main breeding group, either within an enclosure large enough to facilitate meaningful distance from adult birds or via the use of separate, juvenile-only enclosures where birds are able to grow and develop away from adults until they are in good physical condition to re-enter the flock. Such housing could be used for the period immediately after fledging, based on data on wild juvenile dispersal rates and also on age of return to a natal colony [30,32,47], to provide young flamingos with an opportunity to gain a better physical condition before their return to an adult-centric flock where they can develop and refine behaviours essential for adulthood.

Poisson regression analyses highlighted significant differences in nearest neighbour choice at different times of the year. The age class of nearest neighbour had a significant influence on association choice for every age category in both species. It was the only variable to influence the choice of nearest neighbour of Caribbean Flamingos aged 13–24 months in 2014. This result is likely caused by changes in the total number of individuals in each age category fluctuating throughout the year due to the flamingo’s set breeding seasons, with courtship and nesting being dependent on environmental conditions [33,48]. Year and month had a significant effect on the nearest neighbour choices of Greater Flamingos aged 0–12 months and 13–24 months, and for Caribbean Flamingos aged 0–12 months. Once again, this could be caused by flock-wide changes in behaviour caused by breeding. During courtship and nesting periods these juvenile associations may be more regimented because of the lack of involvement of immature birds in adult-specific behaviours (e.g., nesting). The time of day showed a significant effect on nearest neighbour choice for all Greater Flamingo age classes and for Caribbean Flamingos aged 0–12 months. In the youngest of individuals, this could be explained by feeds from parents within the crèche beginning at dusk and carrying on into the night, requiring a potentially large number of adult birds to enter the crèche [26,27]. Similarly, youngsters may crèche together more regularly when parents are away from the breeding area, for feeding, bathing, and preening. However, this does not explain the effect time of day has on individual choice at post-fledging ages, but perhaps the performance of important maintenance activities (e.g., foraging and preening) may cause flamingos to associate more widely across their flock and enclosure, overall.

### 4.2. Research Extensions and Methodological Developments

Our case study could act as a framework for other research projects into the age- and development-related influences on flamingo social choices. Of all species, Flamingos display defined, non-random patterns of social choice when housed in captive flocks [37,39,49], regardless of the overall number of birds present [40]. Therefore, flamingos may judge the quality of potential affiliative (and reproductive) partners from a young age, when spending time with nearest neighbours of specific ages of development. Our results suggest that this captive flock of Caribbean Flamingos did not show the same degree of crèching behaviour as the Greater Flamingo flock. Other factors could have affected the formation and development of the crèche, which in turn caused the juveniles to be in contact with high levels of aggression. Further studies could be conducted on other Caribbean Flamingo flocks and the remaining flamingo species to determine specific differences; recording these changes for a longer period of time would also allow the observer to factor in breeding trends more accurately and better analyse association changes over time. There is also a need for more detailed research on the causation, function, and development of the crèche in captive flocks to support or refute the link between aggression and crèche formation. Further investigation into the changes of association at different times of day and in different parts of the enclosure could prove worthwhile. It would also be interesting to take behaviour (e.g., the action that each flamingo is currently performing) into account and determine changes in association depending on the behaviour being performed. Providing more context to these differences in association (in the form of recording behaviour or calculating any hierarchy or social order within each flock) and considering individual bird social choices by recording each flamingo as an identified individual (where they are in an enclosure and with whom) would remove any influences of pseudoreplication within this dataset caused by the analyses of repeated observations on the same populations of birds.

Key outputs from this paper relate to the conservation of free-living flamingos in natural systems, and to the management of birds under human care. For wild habitat management, margins around the edge of a habitat should be of good quality to allow juveniles to still access quality resources whilst keeping separated from adults. In zoo enclosures, space per bird should be extensive enough to allow congregation of juvenile flamingos of similar age groups away from more aggressive or domineering adults. Zoo population management should consider the size of flamingo colonies so that zoos hold a minimum number of birds that promotes sustainable and regular nesting [50]. Multiple juveniles present per breeding season would allow for the natural development of social behaviours and provide a buffering against stress. Social support [51] from within a group of juvenile birds would be promoted when each flamingo was able to seek out a preferred group of associates. Changes to zoo flamingo enclosures, including the provision of defined nesting and crèching areas, and multiple foraging sites that reduce aggression, would benefit the behavioural development of young flamingos. Previous work has demonstrated that more artificial feeding locations increase aggression between birds and reduce time spent foraging [52], and as juvenile flamingos are less efficient at foraging when compared with adults [10], enclosures should ensure that all birds, regardless of age, can feed without experiencing aggression.

This research should be extended to species of flamingo with an actual and potential conservation need, both in the wild and in captivity, as the results from such studies can help improve wetland habitat structure and resource access for juvenile flamingos as well as evidence-base husbandry and management guidelines for captive care. Breeding flocks of captive Chilean Flamingos and wild flocks of Andean (*Phoenicoparrus andinus*) and Puna (*P. jamesi*) Flamingo may especially benefit from such a focus. This research also has potential applications for the management of the Lesser Flamingo (*Phoeniconaias minor*) too, the regular breeding of which in captive environments has long been considered as challenging [53]. For example, by understanding patterns of juvenile behaviour in the wild, enclosures could be modified accordingly so that when captive Lesser Flamingos do breed, chicks develop in a natural manner.

Given that data were collected in only one location, and on only one flock of each species, we have no replicates of our findings across different flamingo flocks and this limits the wider relevance of our results. Therefore, we encourage others to consider extending such research to add to the generalisability and broader application of our findings to all captive flamingos. Further study should also consider the measurement of individual bird social choice (and enclosure use choice in and around their flock) to provide information on the individual-specific nature of social assortment as flamingos age. Consideration of weather and environmental variables is also important for interpreting captive flamingo behaviour [54], and measuring space use within indoor housing should be considered the next steps for such research, especially if flamingos are housed indoors during periods of inclement weather or due to biosecurity precautions, e.g., Avian Influenza outbreaks [55]. Recording the social choices of known individuals (i.e., via their individual leg rings) would remove any impacts of pseudoreplication that can occur when repeatedly measuring the behaviour of a fixed number of the same animals over time. Such research extensions and further studies would be useful to the development and evolution of husbandry and management guidelines to evidence what young flamingos need as they age and grow, and how to best provide a safe and secure environment that enables successful integration into an adult flock.

## 5. Conclusions

Juvenile Greater and Caribbean Flamingos raised within captive flocks show preferences for associating with youngsters of their own age, and these preferences change over time as they grow and mature. Crèche areas around a nesting site are frequently occupied by the youngest of flamingos, up to 12 months of age, when associating with birds of their own age group. Juvenile flamingos are likely to be seen at the periphery of the main flock, compared with at its centre. Therefore, juvenile flamingos within a captive setting retain an inherent need to remove themselves from the immediate vicinity of adults during their formative years. As flamingos become sub-adults, they re-integrate into the main colony. Therefore, we encourage zoos to consider the behavioural needs of the flamingo chicks hatched by their colonies when planning enclosures and husbandry protocols for these species of bird, which can still cause challenges around sustainable and regular reproduction.

## Figures and Tables

**Figure 1 animals-13-02623-f001:**
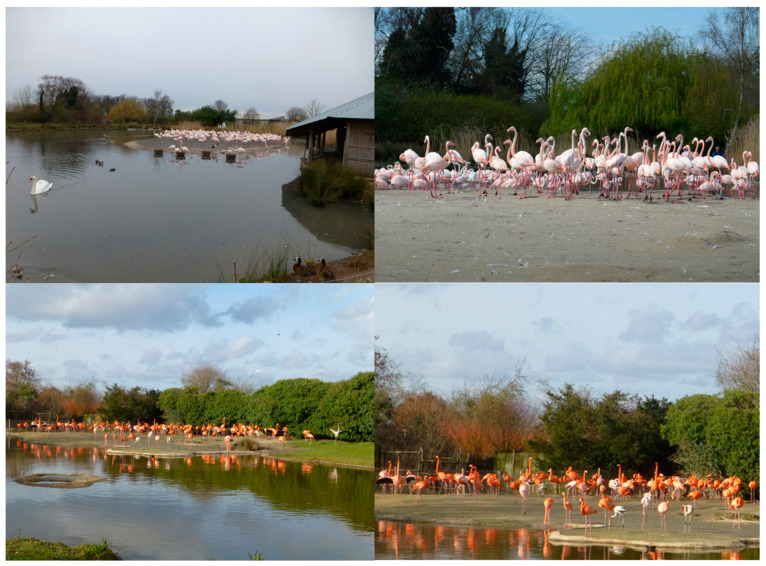
Example of the photographs of each enclosure and each species of flamingo that were used for data collection. Overall flock photos—Greater Flamingos (**top left**) and Caribbean Flamingos (**bottom left**). Closer photos to determine location of juvenile birds compared with adults—Greater Flamingos (**top right**) and Caribbean Flamingos (**bottom right**). Flock periphery was defined as the edge of the main group, with little to no connection to other birds. Flock centre was defined as a juvenile being surrounded by other flamingos in an area that contained the majority of birds in that flock.

**Figure 2 animals-13-02623-f002:**
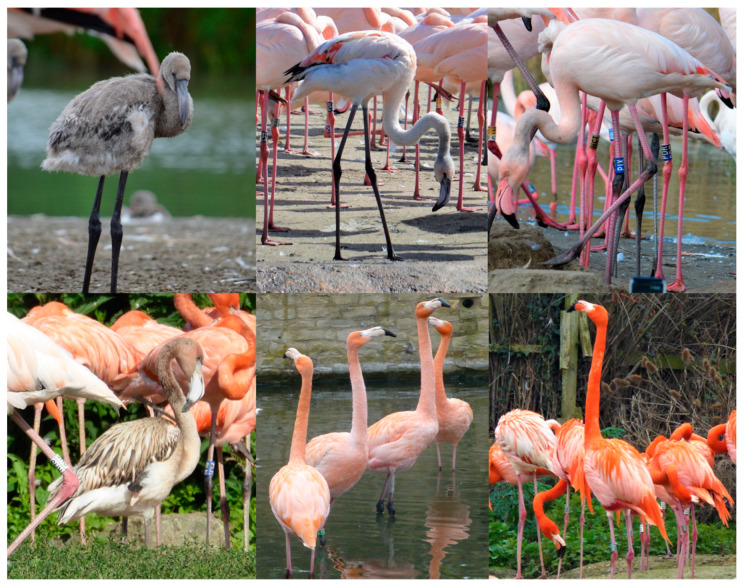
Example of each age category of flamingo (**top row**; Greater Flamingos; **bottom row**, Caribbean Flamingos). From left: up to 12 months; between 13 and 24 months; over 24 months and nearly adult.

**Figure 3 animals-13-02623-f003:**
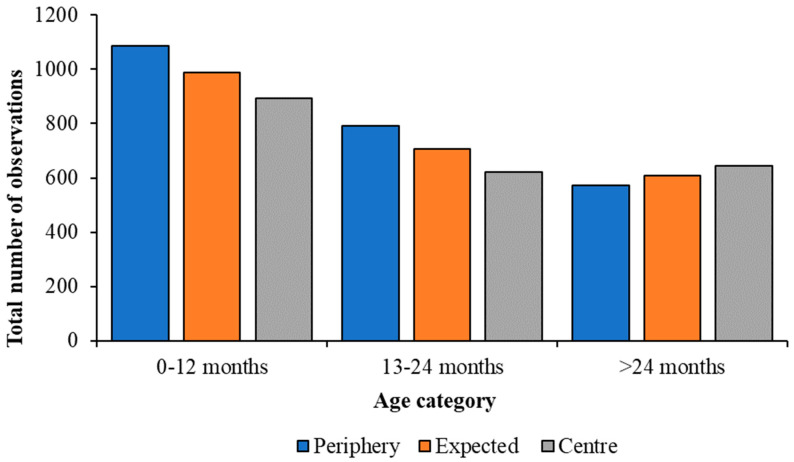
Change in the observed flock position of juvenile Greater Flamingos, compared with an expected value calculated from Chi-squared analyses, for all observations of each age category. Younger birds are seen more at the periphery of the flock and older birds more at the centre.

**Figure 4 animals-13-02623-f004:**
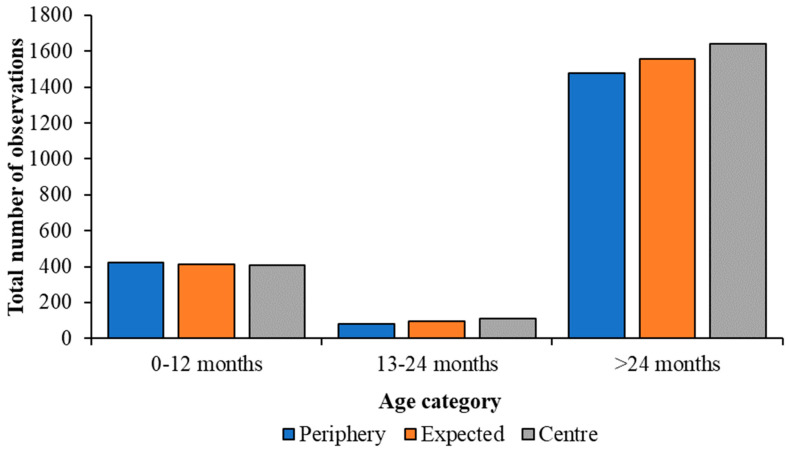
Change in the observed flock position of juvenile Caribbean Flamingos, compared with an expected value calculated from Chi-squared analyses, for all observations of each age category. Older birds are seen more at the centre of the flock.

**Figure 5 animals-13-02623-f005:**
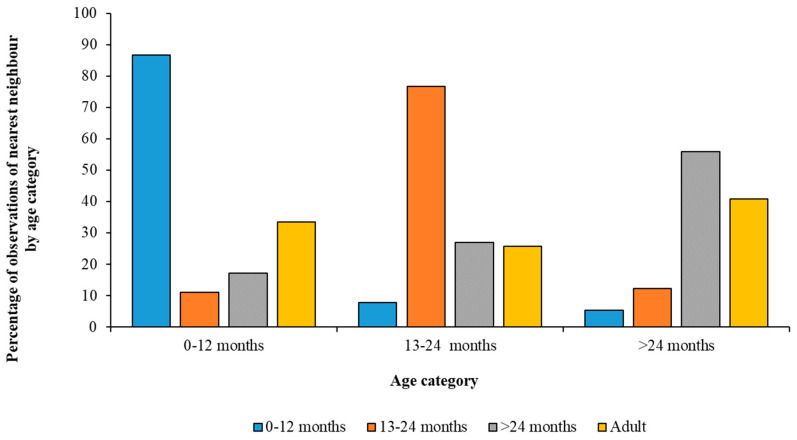
Differences in the age of a flamingo’s nearest neighbour across the different age categories of Greater Flamingos based on the proportion of observed associations. Across all developmental stages, each age category associates most often with birds of their own age.

**Figure 6 animals-13-02623-f006:**
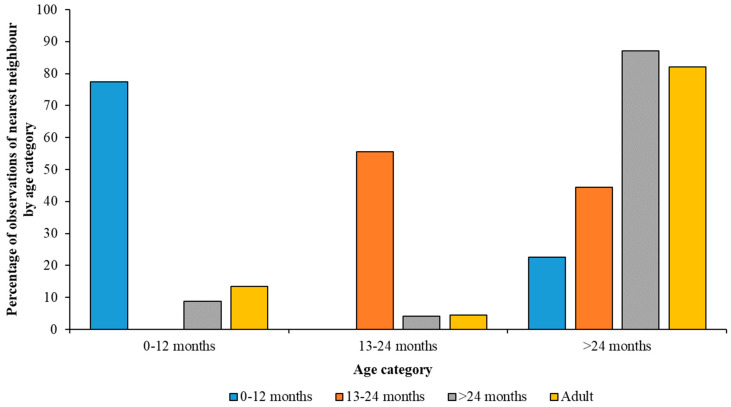
Differences in the age of a flamingo’s nearest neighbour across the different age categories of Caribbean Flamingos based on the proportion of observed associations. Across all developmental stages, each age category associates most often with birds of their own age.

**Figure 7 animals-13-02623-f007:**
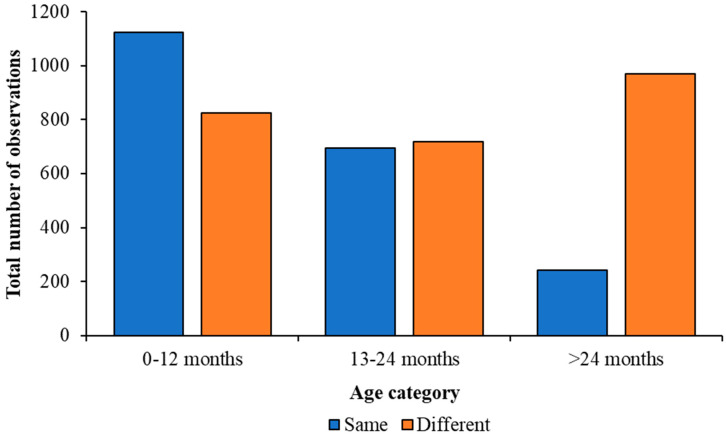
Comparison of the number of times a Greater Flamingo was observed with a nearest neighbour from the same age category as themselves or with a bird from a different age category. Blue bars show observations of a bird associating with the same age category and orange bars show observations with a different age category. Older flamingos were more likely to have a nearest neighbour of an older bird compared with birds in a younger age category.

**Figure 8 animals-13-02623-f008:**
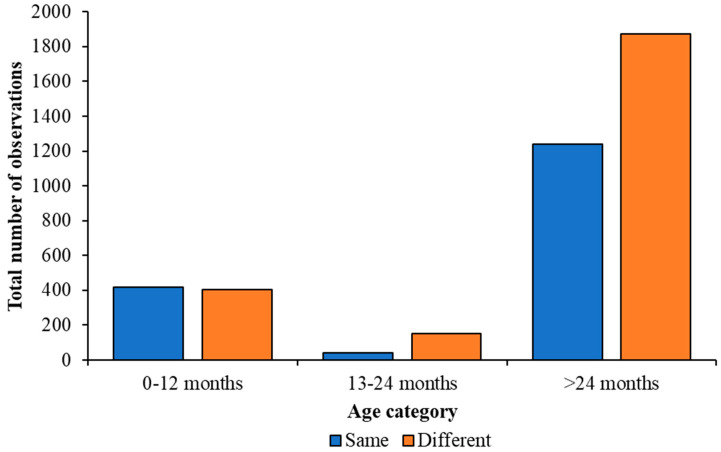
Comparison of the number of times a Caribbean Flamingo was observed with a nearest neighbour from their same age category or with a bird from a different age category. Blue bars show observations of a bird associating with the same age category and orange bars show observations with a different age category. Older flamingos were more likely to have a nearest neighbour of an older bird compared with birds in a younger age category.

**Figure 9 animals-13-02623-f009:**
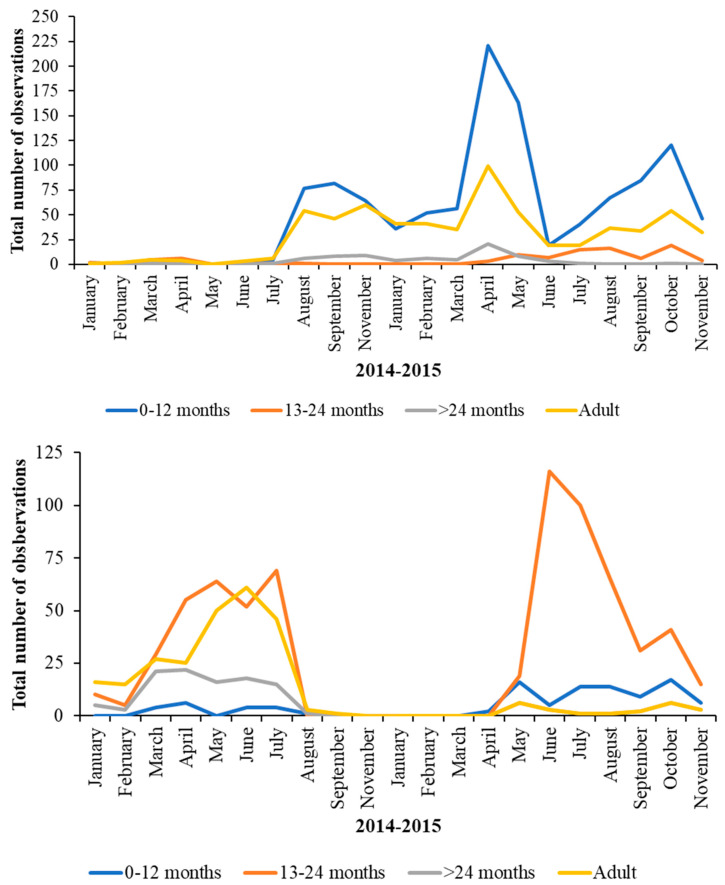
Pattern in the nearest neighbour of Greater Flamingos (**top**) and for Caribbean Flamingos (**bottom**) in the 0–12-month age category over the two years of observations. Both species of flamingo show temporal variation in their choice of nearest neighbour as they age and develop. Younger greater flamingos show defined aggregations with juveniles of their own age in the spring and early summer.

**Figure 10 animals-13-02623-f010:**
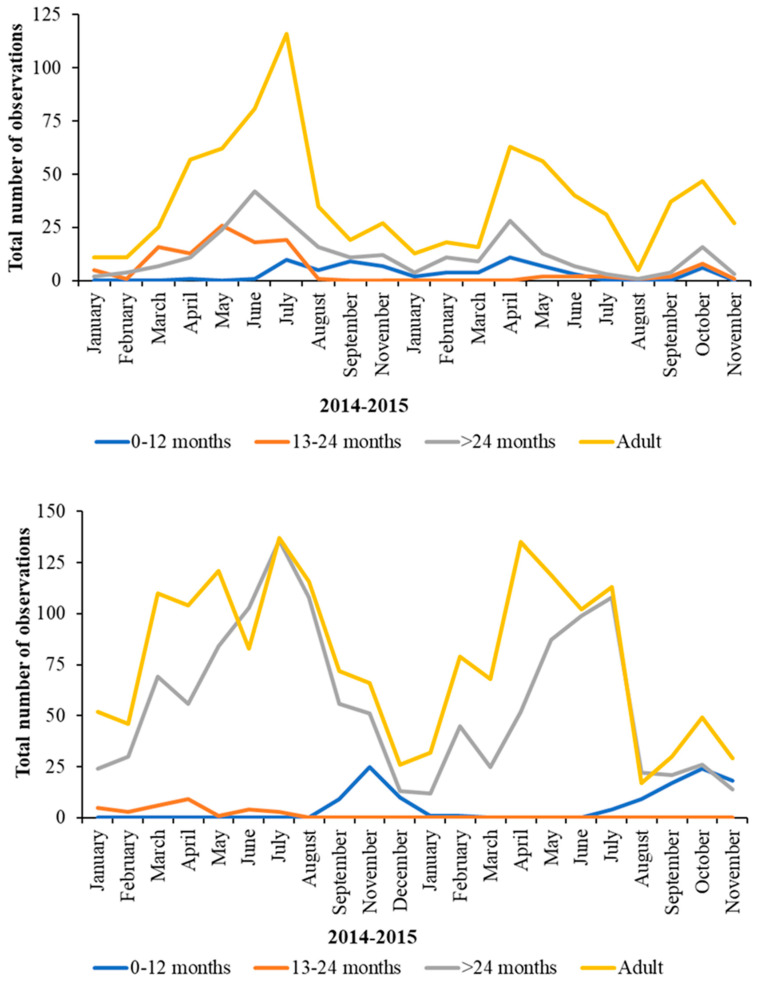
Changes in the age of nearest neighbour for Greater Flamingos aged > 24 months (**top**) and for Caribbean Flamingos aged > 24 months (**bottom**) over the two years of observations. Both species show temporal variation in their choice of nearest neighbour as they age and develop with more observed associations with older flamingos, compared with with younger birds, occurring into the spring and summer months.

**Figure 11 animals-13-02623-f011:**
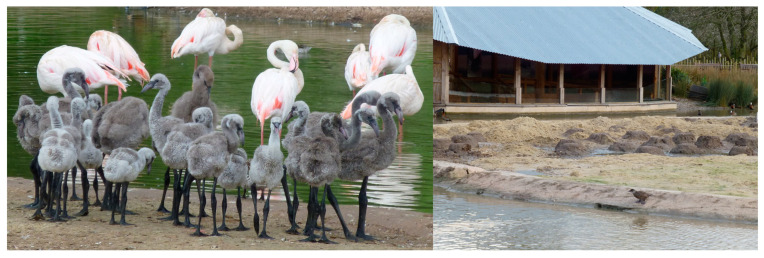
A Greater Flamingo chick crèche (**left**) on the flat, sanded area of a purpose-built nesting island (**right**) that illustrates good practice in providing juvenile flamingos with ecologically relevant enclosure resources to promote behavioural development.

**Table 1 animals-13-02623-t001:** The maximum number of individual flamingos in each age category that were observed in a single photo from across each observation year.

	Greater Flamingo		Caribbean Flamingo	
Age category/Year	2014	2015	2014	2015
0–12 months	16	24	7	17
13–24 months	8	18	4	0
˃24 months	7	11	13	10

**Table 2 animals-13-02623-t002:** Description of zones and their size for each flamingo flock’s enclosure.

Greater Flamingo Enclosure Zones	Zone Size	Caribbean Flamingo Enclosure Zones	Zone Size
Right area of pool	722.5 m^2^ (24%)	Rear nesting island (newer)	24 m^2^ (1.5%)
Back area of pool	675 m^2^ (23%)	Front nesting island (older)	33.6 m^2^ (2.1%)
Outside feeding area in pool	125 m^2^ (4%)	Island near indoor house	23.7 m^2^ (1.5%)
Middle section of pool	236.7 m^2^ (8%)	Outdoor feeding area	13.7 m^2^ (0.9%)
Left area of pool	670.5 m^2^ (23%)	Grass by drain from pool	207 m^2^ (13%)
Crèche area on island	157.1 m^2^ (5%)	Grass by public path	88.2 m^2^ (5.5%)
Nesting area on island	157.1 m^2^ (5%)	Grass under walnut tree	105 m^2^ (6.6%)
Bankside terrestrial areas	224.8 m^2^ (8%)	Grass on right hand side	59 m^2^ (3.7%)
		Sanded area	185 m^2^ (11.6%)
		Back area of water	270.5 m^2^ (17%)
		Middle area of water	173.1 m^2^ (10.9%)
		Front area of water	188 m^2^ (11.8%)
		Water between nesting sites	42.1 m^2^ (2.6%)
		Indoor house	182 m^2^ (11.4%)

**Table 3 animals-13-02623-t003:** The percentage of observations of flamingos from each age category in a central or peripheral position within each enclosure zone. Key out-of-water areas for each enclosure are highlighted with a *.

		0–12 Months		13–24 Months		>24 Months	
Species	Enclosure zone	Centre	Periphery	Centre	Periphery	Centre	Periphery
Greater Flamingo	Feeding zone	3	97	7	93	23	77
	Island *	56	44	61	39	70	30
	Pool	23	77	14	86	22	78
	Land	9	91	0	100	14	86
Caribbean Flamingo	Feeding zone	100	0	71	29	38	62
	Islands	13	88	21	79	14	86
	Pool	38	62	25	75	26	74
	Land *	50	50	67	33	62	38

**Table 4 animals-13-02623-t004:** Count of flamingos of different age categories (and their nearest neighbour’s age category) observed in different enclosure zones for each species.

Age Category		0–12 Months				13–24 Months				>24 Months			
Nearest neighbour age category		0–12	13–24	>24	Ad.	0–12	13–24	>24	Ad.	0–12	13–24	>24	Ad.
Greater Flamingo	Feeding zone	69	2	2	29	3 g	72	13	28	2	12	7	39
	Island	828	71	58	406	77	425	65	308	53	63	186	511
	Pool	161	22	11	160	22	171	45	158	13	42	59	232
	Land	77	0	4	50	0	3	0	7	2	0	5	15
Caribbean Flamingo	Feeding zone	3	Not seen	0	2	Not seen	2	1	4	0	1	22	50
	Island	0	Not seen	2	6	Not seen	6	5	13	2	3	107	177
	Pool	15	Not seen	3	27	Not seen	1	3	8	2	1	161	215
	Land	389	Not seen	118	238	Not seen	31	50	69	114	26	951	1264

## Data Availability

Data can be provided by the corresponding author upon reasonable request.

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
