# Peer review of "Age-Related Change in the Association Choices of Two Species of Juvenile Flamingos"

_animals, 2023, doi:10.3390/ani13162623_

Round 1
Reviewer 1 Report
Dear authors,
This is an interesting study on age-related changes in the association choices of two species of flamingos which can bring new data for zoo management, but also for wild birds’ colonies. However, there are some comments to improve the manuscript.
As a general comment, the authors should decide how they will write the species name. Sometimes they start with capital letter and in other cases no.
Materials and methods
Please add more details on photographs acquisition. How many photos per day/ week / month? Is there any schedule, e.g., 1 in the morning after 2 days 1 in the afternoon? Is there any difference between various time of the day? Maybe they stay together more in the morning and if you made more photos in the first part of the day, this will create a bias in your data. Please clarify the photographs acquisition and explain why you chose this schedule!
Did you check the distribution of your date to see what type of test you must use? Can you give more details about the two-sample proportions test?
Results
As a recommendation, these graphs are not really suitable for a scientific study, maybe you can change them. Also, you can tray to use percent instead of real observation because in those subgroups where you have a high difference in the data is very difficult to understand the differences, e.g. for Caribbean flamingos 80 / 111 observations for 13-24 months, compared to 1475 / 1639 observations for adult birds.
Make the tables and figures caption self-understanding!
Lines 278-298: It will be better to make a table with all these values in order to simplify their presentation. In the text you will explain shortly the tables.
Lines 208 – 329: Check my previous comments; use a table! You can use section 3.5 as an example for the previous two!
In Table 2 you give the area of each enclosure zone, but these areas are not analysed in results and after, in discussions you do not explain their importance. This analysis will be very interesting for species management in captivity or even in the wild.
Discussions
It will be interesting to compare these results with other observation from wild birds to understand if this behaviour is influenced by captivity or this is also happened in wild birds. Also, it will be interesting to have more details about the required space for creches or for adult/subadult groups. This will help Zoo to consider / reconsider their space for these species.
Considering that these birds are captive individuals, is highly important to analyse their behaviour using the current knowledge on wild and captive birds in order to understand if their behaviour is influenced by their captivity.
Minor comments:
The species names are not in the same format, line 239: greater flamingo and line 248: Caribbean flamingos. Please check this in the entire manuscript!
Line 28: Explain what means each abbreviation when you use it for the first time: WWT and UK
Line 92 – 94: There are more species with colonial nesting and creches. Please reformulate this part! You can use these species as examples, but now it looks like these are all bird species with this behavior.
Line 219: What means flock periphery or flock centre? How did you measure these two areas? Did you used a density or a distance, between the individuals / groups?
Lines 267-268: Use the entire species name not only Greater or Caribbean.
Line 463: rOur should be Our…
Best regards,
Author Response
This is an interesting study on age-related changes in the association choices of two species of flamingos which can bring new data for zoo management, but also for wild birds’ colonies. However, there are some comments to improve the manuscript.
As a general comment, the authors should decide how they will write the species name. Sometimes they start with capital letter and in other cases no.
Thank you for the comments and useful information on the paper. We are not sure about the comment on species names. We use a capital for a proper noun and lower case for a regular noun.
Materials and methods
Please add more details on photographs acquisition. How many photos per day/ week / month? Is there any schedule, e.g., 1 in the morning after 2 days 1 in the afternoon? Is there any difference between various time of the day? Maybe they stay together more in the morning and if you made more photos in the first part of the day, this will create a bias in your data. Please clarify the photographs acquisition and explain why you chose this schedule!
We have added in more details to the photo collection schedule.
Did you check the distribution of your date to see what type of test you must use? Can you give more details about the two-sample proportions test?
We have included a statement that data were checked for normality. We are unsure as to what further details the reviewer would like on the proportions test. This test was selected because it compare one choice (same age\0 to another (different) at the population level.
Results
As a recommendation, these graphs are not really suitable for a scientific study, maybe you can change them. Also, you can tray to use percent instead of real observation because in those subgroups where you have a high difference in the data is very difficult to understand the differences, e.g. for Caribbean flamingos 80 / 111 observations for 13-24 months, compared to 1475 / 1639 observations for adult birds.
Thank you for the comment. We have kept in the actual numbers of counts birds associating because those are these data that the proportions test and the Chi squared test used. As we are comparing back to an expected value calculated from the overall number of observations, we are being more valid and open by sharing the counts of data from each age category. If the reviewer can tell us why these graphs are not scientific that would be helpful to their editing, as we have received no other negative comments on them. When graphing data that are used in a Chi squared test it is good practice to show observed and expected values, as we have done here.
Make the tables and figures caption self-understanding!
As above. Please can the reviewer give direction or explanation. Otherwise edits are not possible from limited information. Thank you.
Lines 278-298: It will be better to make a table with all these values in order to simplify their presentation. In the text you will explain shortly the tables.
Thank you for the comment. The descriptive data that could be in a table are illustrated by the bar charts (figures 5 and 6). It would look odd to tabulate the output of inferential testing that supports the differences in these figures. We have moved the figures to the start of the section and placed the inferential testing below to show what is being analysed from the descriptive data.
Lines 208 – 329: Check my previous comments; use a table! You can use section 3.5 as an example for the previous two!
Line 208 is not in the results. This is a huge section. The reviewer would like it all tabulated? Please can we have some clarity.
In Table 2 you give the area of each enclosure zone, but these areas are not analysed in results and after, in discussions you do not explain their importance. This analysis will be very interesting for species management in captivity or even in the wild.
We do not understand what the reviewer means here. We have analysed enclosure occupancy and how it can drive assortment and association. We explain in our methods why we recorded location (line 200) and we analyse this from line 381. We include a table of where flamingos of different ages like to be (table 4). Finally, there is whole section of the discussion (complete with images and recommendations) on enclosure usage- starting at line 441.
Discussions
It will be interesting to compare these results with other observation from wild birds to understand if this behaviour is influenced by captivity or this is also happened in wild birds. Also, it will be interesting to have more details about the required space for creches or for adult/subadult groups. This will help Zoo to consider / reconsider their space for these species.
Considering that these birds are captive individuals, is highly important to analyse their behaviour using the current knowledge on wild and captive birds in order to understand if their behaviour is influenced by their captivity.
Please can the reviewer provide information on the suggested edit or revision? These appear to be statements about the paper’s reach. We have already stated in our discussion that data from wild birds are needed for further validity.
Minor comments:
The species names are not in the same format, line 239: greater flamingo and line 248: Caribbean flamingos. Please check this in the entire manuscript!
Please see our reply above. We give a capital to a proper noun and lower case to a regular noun.
Line 28: Explain what means each abbreviation when you use it for the first time: WWT and UK
We do explain what WWT means in the methods. We have removed the name of the organisation from the abstract.
Line 92 – 94: There are more species with colonial nesting and creches. Please reformulate this part! You can use these species as examples, but now it looks like these are all bird species with this behavior.
We have just used these as examples. We do not state this is an exhaustive list.
Line 219: What means flock periphery or flock centre? How did you measure these two areas? Did you used a density or a distance, between the individuals / groups?
We state in the methods (lines 173 onwards) how the overall flock photo was used to determine where a bird was, based on the juvenile’s location to the main group and their position to other individuals. We also include figure 1 to describe this. Further description has also been included.
Lines 267-268: Use the entire species name not only Greater or Caribbean.
Edited
Line 463: rOur should be Our…
Edited
Reviewer 2 Report
The spatial distribution within groups is usually the result of three factors: resource distributions, predation risk distribution, and intensity of the hierarchical rank. In most gregarious flocks and herds, the dominants get priority access to resources (food sources, mating individuals, etc.) and places with the lowest predation risk (inner locations). In those species with substantial hierarchical differences, subordinate subjects experience delayed access to resources and are located in the group periphery. In most vertebrate species, because the young are the most subordinate subjects, the parents look over them until they grow enough to become independent and climb the hierarchy rank. Some species evolved a sort of communal care, where parents left the chicks in crèches while they forage away from the flock (flamingos, penguins, etc.) or the predation risk increases, and they form anti-predator circles (muskoxen, Wildebeests, Konik ponies). The relative location of subordinates is well-known in many gregarious animals. Because the spatial location is driven by dominance rank, animals tend to interact with their peers, which is another well-known fact. But despite being a known issue, two flamingo species with a mild dominance rank are observed in the present study to explore how the subordinate young are positioned in the group during their first two years of age, compared to subadult and adult birds. The results seem to comply with the factors described above: the young were inside when the flock was disturbed, and they were on the flock's outskirts when they were undisturbed. And young mixed with adults as they got older. There are some concerns about the experimental setup, which cannot provide enough statistical evidence to support the results. They are exposed next.
The same individuals are repeatedly photographed, so an "unreliable" sample size is created by pseudo-replicating the sample of distances with a small sample of birds. Since all individuals must be tagged, the data must be averaged for each tagged individual before statistically analyzing distances between individuals, differences in nearest neighbor ages, preference for being near other individuals of similar ages to its own, and the changes of these parameters throughout the year, etc. Another way to analyze these data is to add a random factor to assume the variance attributable to the individual. The data analyses must control the individual factor.
Dominance ranks were not reported. The distance between birds results from the spatial segregation operated by the dominance rank. But dominance rank was not measured, either as explicitly aggressive or as precautory retreats of the subordinate individual; thus, it cannot be explained the spatial segregation in each flock and flamingo species. Mild dominance ranks must operate in flamingos' spatial distribution. For example, "bumping" or "pushing" interactions has been observed in flamingo that could be a rank proxy. Anyway, an individual measure of dominance rank is needed to explain why the subjects were inner / outer and the distance to other subjects.
Insufficient interspecific and intraspecific experimental replicates. Only one flock of each species was observed. The study must add more flocks, in case of aiming to keep the study of two flamingo species. But since quite a few zoos have other flamingo species, the number of species can also be increased. Conclusion: in its current form, the manuscript does not contain enough sampling units for the study to be biologically sound, even if statistically significant results are obtained.
Minor comments
Page 7, lines 243-246. Results are reported as statistically significant (P<0.05) because the sample is pseudo-replicated. Each subject was recorded repeatedly without including a random factor ('subject') in the statistical analyses. In addition, the differences between the periphery and the centre were minor: 55% vs. 45%, 56% vs. 44%, and 47% vs. 53%, respectively, <12 months, 13-24 months, and subadults. These position differences were 5%, 6%, and 7%, which should be described as small, as expected for a bird species with mild aggressive behaviour and no predation risk for dominant birds kept in zoos. Please comment in the Discussion on the small effect size of this frequency comparison. The result is perhaps valuable because low spatial segregation is expected in mild hierarchical species.
Typos
Page 13, Figure 9. Change '2014 to 2015' with '2014-2015' in the x-axis.
Author Response
The spatial distribution within groups is usually the result of three factors: resource distributions, predation risk distribution, and intensity of the hierarchical rank. In most gregarious flocks and herds, the dominants get priority access to resources (food sources, mating individuals, etc.) and places with the lowest predation risk (inner locations). In those species with substantial hierarchical differences, subordinate subjects experience delayed access to resources and are located in the group periphery. In most vertebrate species, because the young are the most subordinate subjects, the parents look over them until they grow enough to become independent and climb the hierarchy rank. Some species evolved a sort of communal care, where parents left the chicks in crèches while they forage away from the flock (flamingos, penguins, etc.) or the predation risk increases, and they form anti-predator circles (muskoxen, Wildebeests, Konik ponies). The relative location of subordinates is well-known in many gregarious animals. Because the spatial location is driven by dominance rank, animals tend to interact with their peers, which is another well-known fact. But despite being a known issue, two flamingo species with a mild dominance rank are observed in the present study to explore how the subordinate young are positioned in the group during their first two years of age, compared to subadult and adult birds. The results seem to comply with the factors described above: the young were inside when the flock was disturbed, and they were on the flock's outskirts when they were undisturbed. And young mixed with adults as they got older. There are some concerns about the experimental setup, which cannot provide enough statistical evidence to support the results. They are exposed next.
Thank you for the comments. The reviewer suggests that flamingos were disturbed but this did not occur, nor did we measure disturbance or artificially make the birds disturbed to gain a response. We have simply counted where we have seen juvenile birds at different time points, to see if there is any relationship between age (based on plumage colour) and location in the enclosure / location to the rest of the flock.
The same individuals are repeatedly photographed, so an "unreliable" sample size is created by pseudo-replicating the sample of distances with a small sample of birds. Since all individuals must be tagged, the data must be averaged for each tagged individual before statistically analyzing distances between individuals, differences in nearest neighbor ages, preference for being near other individuals of similar ages to its own, and the changes of these parameters throughout the year, etc. Another way to analyze these data is to add a random factor to assume the variance attributable to the individual. The data analyses must control the individual factor.
Thank you for the comments and feedback. We are afraid that we do not know how to fully respond to this review because the comments seem to be asking for a paper that we haven’t written. We have described where we have seen flamingos of different age categories during development. We have discussed differences in adult and juvenile behaviour, specifically aggression, that may cause juveniles to be in different areas of the enclosure. We have suggested ways that zoos can alter their enclosures to make peripheral areas of the enclosure better for juveniles. We haven’t made predictions on individual factors that may cause birds to be in different places in an enclosure based on dominance or hierarchy.
Dominance ranks were not reported. The distance between birds results from the spatial segregation operated by the dominance rank. But dominance rank was not measured, either as explicitly aggressive or as precautory retreats of the subordinate individual; thus, it cannot be explained the spatial segregation in each flock and flamingo species. Mild dominance ranks must operate in flamingos' spatial distribution. For example, "bumping" or "pushing" interactions has been observed in flamingo that could be a rank proxy. Anyway, an individual measure of dominance rank is needed to explain why the subjects were inner / outer and the distance to other subjects.
This paper has no aims to measure or calculate dominance rank. We did not measure these factors because flamingos are an obligate colonial species that, accordingly, have poor resource defence and no linear dominance hierarchy or otherwise. The only research on dominance in flamingos has been undertaken in small captive flocks with limited replication and is not encountered in the wild. We never intended to measure dominance. Simply to look at where do we see juveniles within the group. Are they on the outskirts of the flock, or are they more central? The reviewer has suggested a refocus and reframing of the paper that is not linked to our project nor what we aim to show. We are providing zoos with information on where captive flamingos, of different age may (or may not) like to be, and why this is important for bird welfare and development.
Insufficient interspecific and intraspecific experimental replicates. Only one flock of each species was observed. The study must add more flocks, in case of aiming to keep the study of two flamingo species. But since quite a few zoos have other flamingo species, the number of species can also be increased. Conclusion: in its current form, the manuscript does not contain enough sampling units for the study to be biologically sound, even if statistically significant results are obtained.
Thank you for the comments. We do not have replicates nor did we set out with an intention of writing a paper across flocks and across different species. We had access to photographic data for two flocks of birds at one institution that contained juveniles and adults. We used these photos for this case study to identify trends and patterns in where juveniles are compared to adults. The reviewer is, again, asking for a paper that is not what we have created. It is not a multi zoo, multi species study of flamingo dominance or individual differences in flock hierarchy. This paper describes where young birds have been seen why this may be important for zoos to advance welfare and husbandry. We are sorry that the reviewer feels the paper is both biologically and scientifically unsound but we respectively disagree and feel it has value to advancing flamingo care in zoos.
Minor comments
Page 7, lines 243-246. Results are reported as statistically significant (P<0.05) because the sample is pseudo-replicated. Each subject was recorded repeatedly without including a random factor ('subject') in the statistical analyses. In addition, the differences between the periphery and the centre were minor: 55% vs. 45%, 56% vs. 44%, and 47% vs. 53%, respectively, <12 months, 13-24 months, and subadults. These position differences were 5%, 6%, and 7%, which should be described as small, as expected for a bird species with mild aggressive behaviour and no predation risk for dominant birds kept in zoos. Please comment in the Discussion on the small effect size of this frequency comparison. The result is perhaps valuable because low spatial segregation is expected in mild hierarchical species.
We have edited the discussion to include discussion of potential pseudoreplication.
We have clearly stated in our discussion that this research needs to be extended, and that it could act as a framework for further research. We feel that the comments on pseudo-replication, when we have counted across a large dataset (nearly 9000), are unfair as it is unlikely that we would be always seeing the same bird in the same position. The reason we have chosen to run “basic” inferential testing is because of the case study approach; to not try and predict why patterns in association or location exist, but to describe what we observed. As this is more likely to be useful for practitioners and animal care staff. Again, we do not have a paper on individual bird preferences, but what the population is doing at different sample points. We haven’t analysed individual differences because it was not possible to always identify birds from photos, not all birds were ringed and we were interested in general trends. We have already commented in the discussion that replication of this research should occur across more flocks and at different institutions and in the wild.
Typos
Page 13, Figure 9. Change '2014 to 2015' with '2014-2015' in the x-axis.
Edited
Reviewer 3 Report
Dear authors, thank you very much for the opportunity to read such an interesting study. This study is very relevant not also to understanding flamingo biology but for the management of the species. I made a few recommendations on the pdf file. Please, verify. Best regards

Author Response
We thank the reviewer for the helpful and developmental comments on the manuscript.
We have corrected the manuscript directly based on all other suggestions.
We will edit figure 3 to include where data are significant.
In response to the to question on Table 1, this is just to show the maximum number of juveniles that were seen in a photo across data collection. It does not represent the total overall number of juveniles in each flock.
The reviewer’s interpretation of Figure 9 is correct.
Reviewer 4 Report
The paper on association choices in juvenile flamingo species is an interesting and valuable piece of research.
The work is generally well written and easy to follow and the methods are robust and sound.
I have made some suggestions for better presentation and analysis for the authors to improve the work.
Line 55/56 this is not true for all vertebrates, please reword this sentence.
Line 58/59 needs citation
Line 72/73 which study? Assume previous sentence but could be referring to the present study.
Line 117 is this true for all species of flamingo?
A well written and comprehensive introduction, though additional information on the difference in breeding behaviors and development between the two species could have been included in the introduction.
Methods – were the images taken every day, or if not what was the max and min time gap between photos. As time of day had an effect on proximity of different age classes the specifics of when these photos were taken should be made clear.
Line 200-201 is this sentence unfinished – juveniles were recorded as being in the center of the group?
Was there classification difference in periphery (close to the edge) and outside the main group body?
Table two – typo in the measurements mh2?
Nearest neighbor mentioned in the data analysis but this has not been explained as to how the nearest neighbor was calculated, measured distance, nominally noted. Based on straight line distance or split by enclosure feature. What was used when multiple nearest neighbors?
Line 230 interacting with what?
Line 234 and figure 3, has this been adapted by proportions? You have more 0-12 month individuals in table 1 and therefore a greater number of observations would be expected here.
What are the expected values based on?
Would a two factor chi square test not work better here to compare the different age categories rather than just comparing with equal placement within age categories?
Again figure 5 and 6 seem misleading by showing total number of observations, equally the Chi square test should consider the proportion of available observations with other age categories as nearest neighbor.
Given that you have used proportional testing in section 3.3 the figures should also reflect this.
Line 341 and 354 how does time of day affect aggregation of juveniles and adults?
Line 480 would the inclusion of a separate juvenile area not prevent learnt breeding behaviors from the adults and opportunities for juveniles to disperse back to the center of the flock when they feel ready? Given this difference between the species observed, it would be worth noting that this may be detrimental and particularly so for species that show less juvenile separation?
Author Response
The paper on association choices in juvenile flamingo species is an interesting and valuable piece of research. The work is generally well written and easy to follow and the methods are robust and sound. I have made some suggestions for better presentation and analysis for the authors to improve the work.
Thank you for the comments and we are pleased that you have found the manuscript useful and informative for zoos.
Line 55/56 this is not true for all vertebrates, please reword this sentence.
You are indeed correct! Apologies for this typo. All has become many.
Line 58/59 needs citation
A new citation has been provided.
Line 72/73 which study? Assume previous sentence but could be referring to the present study.
Edited
Line 117 is this true for all species of flamingo?
Yes, all flamingo species appear to fledge after a similar amount of time and hatch after the same incubation period.
A well written and comprehensive introduction, though additional information on the difference in breeding behaviors and development between the two species could have been included in the introduction.
Thank you for the comments. We did not include more information on the flamingos themselves their courtship display, nest building and chick rearing is the same across species. But we have included a sentence in the method stating the similar ecology of these two species (that we considered subspecies until 2002).
Methods – were the images taken every day, or if not what was the max and min time gap between photos. As time of day had an effect on proximity of different age classes the specifics of when these photos were taken should be made clear.
Thank you for the comment. Time of day has been included in the methods.
Line 200-201 is this sentence unfinished – juveniles were recorded as being in the center of the group?
Apologies for the typo. This has been corrected.
Was there classification difference in periphery (close to the edge) and outside the main group body?
Thank you for the comment. We have attempted to further describe this in the methods themselves as well as in the caption for Figure 1.
Table two – typo in the measurements mh2?
Edited. Thank you for spotting.
Nearest neighbor mentioned in the data analysis but this has not been explained as to how the nearest neighbor was calculated, measured distance, nominally noted. Based on straight line distance or split by enclosure feature. What was used when multiple nearest neighbors?
Apologies for the oversight. This information was included in a draft but seemed to have not been copied onto the MDPI template. This information is included under Figure 2.
Line 230 interacting with what?
This has been edited to “associating”.
Line 234 and figure 3, has this been adapted by proportions? You have more 0-12 month individuals in table 1 and therefore a greater number of observations would be expected here.
Thank you for the comment. The information in Table 1 shows the maximum number of juveniles seen at any one point in a photo. Because juveniles aged across the course of the study we have included the total number of observations into the chi squared testing (that these graphs are based on) to compare the observed value against an expected value calculated from all observations of when a juvenile flamingo (within each age category) was recorded.
What are the expected values based on?
The expected values were calculated by Minitab automatically based on the observational data inputted into the Chi squared testing.
Would a two factor chi square test not work better here to compare the different age categories rather than just comparing with equal placement within age categories?
Thank you for the comment. We used a one way Chi squared to compare within the age categories themselves.
Again figure 5 and 6 seem misleading by showing total number of observations, equally the Chi square test should consider the proportion of available observations with other age categories as nearest neighbor. Given that you have used proportional testing in section 3.3 the figures should also reflect this.
Thank you for the comments. We have edited figures 5 and 6 to show proportions off observations (as a percentage)
Line 341 and 354 how does time of day affect aggregation of juveniles and adults?
Edited to include the effect of time as a predictor on associations.
Line 480 would the inclusion of a separate juvenile area not prevent learnt breeding behaviors from the adults and opportunities for juveniles to disperse back to the center of the flock when they feel ready? Given this difference between the species observed, it would be worth noting that this may be detrimental and particularly so for species that show less juvenile separation?
Thank you for the useful and relevant point. We have expanded on this to clarify our idea.
Round 2
Reviewer 1 Report
Dear authors,
The manuscript has been improved, but there are some aspects that need clarification. In scientific communication, there are specific rules when using common names, especially in a scientific paper. The species name should always be written with a capital letter, even if it is a regular noun, as it represents the species itself. You can refer to taxonomic sites to verify these writing practices.
Materials and methods:
I noticed the photo collection schedule, which is now clearer regarding the methodology. However, I have some questions. If you took photos four times per day over two years, this would result in almost 2920 photos, but you only used 825. What happened to the other photos? Are there any specific rules for selecting certain photos or time points? Additionally, please include the results of the Kolmogorov-Smirnov test to clarify the distribution of your data.
Results:
It would be beneficial to present the graphs in a more scientifically appropriate design. While large and colorful graphs may be suitable for public presentations, scientific papers require a more standard and objective design.
Ensure that the table and figure captions are self-explanatory. A reader should be able to understand the content of the table or graph without reading the main text. Therefore, include all the necessary details in the caption to convey the key findings presented in the table or figure.
Regarding the zones, I meant to inquire about how the different zones within the enclosure were used. You have mentioned these zones in Table 2.
Discussions:
While some publications have been used, I encourage the authors to explore more scientific literature during the writing process, as thorough documentation is a fundamental requirement for scientific articles.
https://www.wetlands.org/wp-content/uploads/2015/11/Flamingo-Special-Publication-1.pdf
https://www.sciencedirect.com/science/article/pii/S0168159118302430
Please address these points to enhance the clarity and scientific rigor of your manuscript.
Best regards,
Author Response
Dear authors,
The manuscript has been improved, but there are some aspects that need clarification. In scientific communication, there are specific rules when using common names, especially in a scientific paper. The species name should always be written with a capital letter, even if it is a regular noun, as it represents the species itself. You can refer to taxonomic sites to verify these writing practices.
Thank you for the comment and feedback. We am afraid that we don’t agree that bird names have to always be written in capital letters. We have published numerous papers where the common name is lower case unless a proper noun. However, we have edited all common names to have capital letters in this case.
Materials and methods:
I noticed the photo collection schedule, which is now clearer regarding the methodology. However, I have some questions. If you took photos four times per day over two years, this would result in almost 2920 photos, but you only used 825. What happened to the other photos? Are there any specific rules for selecting certain photos or time points? Additionally, please include the results of the Kolmogorov-Smirnov test to clarify the distribution of your data.
We do not understand how the reviewer has worked out this calculation of photo number? The same number of photos per flock were not always taken per day, as this depended on the behaviour of the flock within the enclosure, and we were using a sample of photos collected from a previous project that was identifying social bonds between birds. We have referenced back to this project in the text to show what the original photo dataset was collected for and the methods applied. And we have clearly stated that only photos that included juveniles were used for this article. Please see the edit.
The result from the Kolmogorov-Smirnov test was simply <0.05 suggesting that data were not normally distributed. Had data been normally distributed, the P value from this test would have been >0.05. The text has been edited.
Results:
It would be beneficial to present the graphs in a more scientifically appropriate design. While large and colorful graphs may be suitable for public presentations, scientific papers require a more standard and objective design.
Thank you for the comment. This journal allows for full colour online publishing of illustrations, figures and charts. Colour graphs are now common in scientific journals and can provide further clarity of results. We respectfully wish to keep these figures as is, as we have been consistent with style throughout.
Ensure that the table and figure captions are self-explanatory. A reader should be able to understand the content of the table or graph without reading the main text. Therefore, include all the necessary details in the caption to convey the key findings presented in the table or figure.
We have attempted to edit captions and figures were needed but we are still unsure as to what tables and figures are not well described. All figures and tables are linked back to analyses presented in the text as is the norm for a scientific paper.
Regarding the zones, I meant to inquire about how the different zones within the enclosure were used. You have mentioned these zones in Table 2.
Thank you for the comment. These zones were defined on resources accessible to the birds, and again this was based on previous study that has already been referenced in the article. However, we have provided further clarity.
Discussions:
While some publications have been used, I encourage the authors to explore more scientific literature during the writing process, as thorough documentation is a fundamental requirement for scientific articles.
Thank you for the comment, but we do not understand what the revision is here. We have used a wide range of references in the discussion to support and extend our points. We would gladly include more references if specific points of discussion were highlighted for review.
https://www.wetlands.org/wp-content/uploads/2015/11/Flamingo-Special-Publication-1.pdf
This is an entire newsletter. Which part of it should we take specifically to enhance the discussion? There do not appear to be any papers in it that relate to flamingo social assortment?
https://www.sciencedirect.com/science/article/pii/S0168159118302430
This links back to a paper that is already referenced in the manuscript. And is a paper from one of the authors of this article so we have been careful to read widely and not over-reference our own citations. Please provide more details.
Please address these points to enhance the clarity and scientific rigor of your manuscript.
We hope that we have done so.
Reviewer 2 Report
The revised manuscript has improved as far as the experimental design, statistical analyzes, and data from just two repeatedly observed flocks of flamingos allows. I have no new suggestions to offer to improve this manuscript.
Author Response
Thank you for your comments on the manuscript. It has been helpful in developing this paper further.